# Effective Higher-order Link Prediction and Reconstruction from Simplicial Complex Embeddings

**Simone Piaggesi**
University of Bologna, Italy
ISI Foundation, Torino, Italy
simone.piaggesi2@unibo.it

**André Panisson**
CENTAI, Torino, Italy
andre.panisson@centai.eu

**Giovanni Petri**
CENTAI, Torino, Italy
giovanni.petri@centai.eu

## Abstract

Methods that learn graph topological representations are becoming the usual choice to extract features to help solve machine learning tasks on graphs. In particular, low-dimensional encoding of graph nodes can be exploited in tasks such as link prediction and network reconstruction, where pairwise node embedding similarity is interpreted as the likelihood of an edge incidence. The presence of polyadic interactions in many real-world complex systems is leading to the emergence of representation learning techniques able to describe systems that include such polyadic relations. Despite this, their application on estimating the likelihood of tuple-wise edges is still underexplored.

Here we focus on the reconstruction and prediction of simplices (higher-order links) in the form of classification tasks, where the likelihood of interacting groups is computed from the embedding features of a simplicial complex. Using similarity scores based on geometric properties of the learned metric space, we show how the resulting node-level and group-level feature embeddings are beneficial to predict unseen simplices, as well as to reconstruct the topology of the original simplicial structure, even when training data contain only records of lower-order simplices.

## 1 Introduction

Network science provides the dominant paradigm for the study of the structure and dynamics of complex systems, thanks to its focus on their underlying relational properties. In data mining applications, topological node embeddings of networks are standard representation learning methods that help solve downstream tasks, such as network reconstruction, link prediction, and node classification [1]. Complex interacting systems have been usually represented as graphs. This representation however suffers from the obvious limitation that it can only capture pairwise relations among nodes, while many systems are characterized by group interactions [2]. Indeed, simplicial complexes are generalized graphs that encode group-wise edges as sets of nodes, or *simplices*, with the additional requirement that any subset of nodes forming a simplex must also form a simplex belonging to the complex. Unlike alternative high-order representations, e.g. hypergraphs, which also overcome the dyadic limitation of the graph formalism [3], the simplicial *downward closure* constraint works particularly well when studying systems with subset dependencies, such as brain networks and social networks (e.g., people interacting as a group also engage in pairwise interactions).

Due to the increased interest in studying complex systems as generalized graph structures, topological representation learning techniques on simplicial complexes are also emerging as tools to solve learning tasks on systems with polyadic relations. In particular, here we focus on tasks based on the reconstruction and prediction of higher-order edges. While for standard graphs these problems have been extensively studied with traditional machine learning approaches [4, 5] and representation learning [6, 7], the literature for their higher-order counterparts is more limited. In fact, reconstruction and prediction of higher-order interactions have been investigated mainly starting from pairwise data [8, 9] or time series [10, 11], without particular attention to representation learning methods.

S. Piaggesi, A. Panisson, G. Petri, Effective Higher-order Link Prediction and Reconstruction from Simplicial Complex Embeddings. *Proceedings of the First Learning on Graphs Conference (LoG 2022)*, PMLR 198, Virtual Event, December 9–12, 2022.

Here we study low-dimensional embeddings of simplicial complexes for link prediction and reconstruction in higher-order networks. Our main contributions are:

- We introduce an embedding framework to compute low-rank representations of simplicial complexes.
- We formalize network reconstruction and link prediction tasks for polyadic graph structures.
- We show that simplicial similarities computed from embedding representations outperform classical network-based reconstruction and link prediction methods.

Since the problems of link prediction and network reconstruction are not yet well-defined in the literature for the higher-order case, none of the available state-of-the-art methods were previously evaluated in terms of both these tasks. In this paper, we properly delineate the formal steps to perform higher-order link prediction and reconstruction, and we make a comprehensive evaluation of different methods adding many variations such as the use of multi-node proximities and simplicial weighted random walks. We publicly release the code to run the experiments at `https://github.com/simonepiaggesi/simplex2pred`.

## 2 Related Work

***Representation Learning Beyond Graphs.*** Representation learning for graphs [1] allows obtaining low-dimensional vector representations of nodes that convey information useful for solving machine learning tasks. Most methods fit into one of these two categories: *shallow node embeddings* and *graph neural networks* (GNNs). Shallow methods generate node representations as a result of an unsupervised task (e.g., matrix factorization [12]), while GNN methods obtain node vectors from iterative message passing operations, e.g. graph convolutions and graph attention networks [13]. In hypergraph settings, node embedding methods typically leverage hyperedge relations similarly to what is done for standard graph edges: for example, spectral decomposition [14], random walk sampling [15, 16], autoencoders [17]. Recently, Maleki et al. [18] proposed a hierarchical approach for scalable node embedding in hypergraphs. In simplicial complexes, random walks over simplices are exploited to compute embeddings of interacting groups with uniform or mixed sizes [19, 20], extending hypergraph methods that compute only node representations. Extensions of GNNs have been proposed to generalize convolution and attention mechanisms to hypergraphs [21–24] and simplicial complexes [25–27].

***Link Prediction and Network Reconstruction Beyond Graphs.*** The *link prediction* [4] task predicts the presence of unobserved links in a graph by estimating their occurrence likelihood, while *network reconstruction* consists in the inference of a graph structure based on indirect data [28], missing or noisy observations [29]. In this work, we use latent embedding variables to assess the reconstruction and prediction of a given edge, relying on similarity indices. In higher-order systems, link prediction has been investigated primarily for hypergraphs, in particular with methods based on matrix factorization [30, 31], resource allocation metric [32], loop structure [33], and representation learning [34, 35]. The higher-order link prediction problem was introduced in a temporal setting by Benson et al. [9] (reformulating the term *simplicial closure* [36]), while Liu et al. [37] studied the prediction of several higher-order patterns with neural networks. Yoon et al. [38] investigated the use of opportune $k$-order projected graphs to represent group interactions, and Patil et al. [39] analyzed the problem of finding relevant candidate hyperlinks as negative examples. Despite these early results, reconstruction of higher-order interactions is an ongoing challenge: for example, Young et al. [8] proposed a Bayesian inference method to distinguish between hyperedges and combinations of low-order edges in pairwise data, while Musciotto et al. [40] developed a filtering approach to detect statistically significant hyperlinks in hypergraph data. In addition, some works studied approaches for the inference of higher-order structures from time series data [10, 11].

## 3 Methods and Tasks Description

### 3.1 Reconstruction and Prediction of Higher-order Interactions in Simplicial Complexes

Simplicial complexes can be considered as generalized graphs that include higher-order interactions. Given a set of nodes $\mathcal{V}$, a simplicial complex $\mathcal{K}$ is a collection of subsets of $\mathcal{V}$, called *simplices*, satisfying *downward closure*: for any simplex $\sigma \in \mathcal{K}$, any other simplex $\tau$ which is a subset of $\sigma$

belongs to the simplicial complex $\mathcal{K}$ (for any $\sigma \in \mathcal{K}$ and $\tau \subset \sigma$, we also have $\tau \in \mathcal{K}$). This constraint makes simplicial complexes different from *hypergraphs*, for which there is no prescribed relation between hyper-edges. A simplex $\sigma$ is called a $k$-simplex if $|\sigma| = k + 1$, where $k$ is its *dimension* (or order). A simplex $\sigma$ is a *coface* of $\tau$ (or equivalently, $\tau$ is a *face* of $\sigma$) if $\tau \subset \sigma$. We denote with $\dim(\sigma)$ the order of simplex $\sigma$, and with $n_k$ the number of $k$-simplices in $\mathcal{K}$.

Given a simplicial complex $\mathcal{K}$, by *reconstruction* of higher-order interactions we mean the task of correctly classifying whether a group of $k + 1$ nodes $s = (i_0, i_1, \ldots, i_k)$ is a $k$-simplex of $\mathcal{K}$ or not. More specifically, we consider $\mathcal{S} = \{ s \in \mathcal{K} : |s| > 1 \}$ as the set of interactions (simplices with order greater than 0) that belongs to the simplicial complex $\mathcal{K}$. Given any group $s = (i_0, i_1, \ldots, i_k)$, with the reconstruction task we aim to discern if the elements in $s$ interact within the same simplex, and so $s \in \mathcal{S}$, or $s$ is a group of lower-order simplices, and so $s \notin \mathcal{S}$ (but subsets of $s$ may be existing simplices). When group $s$ interacts within a simplex, we say that $s$ is *closed*, conversely it is *open*.

By higher-order interaction *prediction* we mean instead the task of predicting whether an interaction $\mathcal{S}^*$ that has not been observed at a certain time (i.e., the simplex has not been added to the complex yet) will appear in the future. Given any open configuration $\bar{s} \in \mathcal{U}_\mathcal{S}$ coming from the set of unobserved interactions $\mathcal{U}_\mathcal{S} = \{ s \in 2^\mathcal{V} : |s| > 1, s \notin \mathcal{S} \}$, namely the complement[1] of $\mathcal{S}$, the prediction task is to classify which groups will give rise to a simplicial closure in the future ($\bar{s} \in \mathcal{S}^*$) versus those that will remain open ($\bar{s} \in \mathcal{U}_\mathcal{S} \setminus \mathcal{S}^*$).

## 3.2 Low-dimensional Embedding of Simplicial Complexes

Given a simplicial complex $\mathcal{K}$, we want to learn a mapping function $f : \mathcal{K} \to \mathbb{R}^d$ from elements of $\mathcal{K}$ to a $d$-dimensional low-rank feature space ($d \ll |\mathcal{K}|$). The mapping $f$ must preserve topological information incorporated in the simplicial complex, in such a way that adjacency relations are preserved into geometric distances between vectors of the embedding space. Here we propose that representations of simplices can be obtained by random-walking over the inclusions hierarchy of $\mathcal{K}$ and learning the embedding space according to the simplex proximity observed through such walks, preserving high-order information about the topological structure of the complex itself.

The navigation of the downward inclusion chain can be performed with usual graph random walk sampling, unfolding the simplicial complex in its canonical graph of inclusions, called Hasse Diagram (HD): formally, the Hasse Diagram $\mathcal{H}(\mathcal{K})$ of complex $\mathcal{K}$ is the multipartite graph $\mathcal{H}(\mathcal{K}) = (\mathcal{V}_\mathcal{H}, \mathcal{E}_\mathcal{H})$, such that each node $v_\sigma \in \mathcal{V}_\mathcal{H}$ corresponds to a simplex $\sigma \in \mathcal{K}$, and two simplices $\sigma, \tau \in \mathcal{K}$ are connected by the undirected edge $(v_\sigma, v_\tau) \in \mathcal{E}_\mathcal{H}$ iff $\sigma$ is a coface of $\tau$ and $\dim(\tau) = \dim(\sigma) - 1$. In other words, each simplicial order corresponds to a graph layer in $\mathcal{H}(\mathcal{K})$, and two simplices in different layers are linked if they are (upper/lower) adjacent in the original simplicial complex. The optimization problem defined here is independent of the random walk sampling procedure, so in our experiments we test different procedures (listed in §4).

Inspired by language models such as WORD2VEC [41], we start from a corpus $\mathcal{W} = \{ \sigma_1, \ldots, \sigma_{|\mathcal{W}|} \}$ of simplicial random walks, and we aim to maximize the log-likelihood of a target simplex $\sigma_i$ given the multi-set $\mathcal{C}_T(\sigma_i) = \{ \sigma_{i-T} \ldots \sigma_{i-1}, \sigma_{i+1} \ldots \sigma_{i+T} \}$ of context simplices within a distance $T$, determined as the number of steps between the target and the context simplex. The optimization problem is as follows:

$$\max_f \ \sum_{i=1}^{|\mathcal{W}|} \log \Pr( \sigma_i \mid \{ f(\tau) : \tau \in \mathcal{C}_T(\sigma_i) \} ) \tag{1}$$

where the probability is the soft-max $\Pr(\sigma_i \mid \{ f(\tau), \ldots \}) \propto \exp \left[ \sum_{\tau \in \mathcal{C}_T(\sigma_i)} f(\sigma_i) \cdot f(\tau) \right]$, normalized via the standard partition function $Z_{\sigma_i} = \sum_{\kappa \in \mathcal{K}} \exp \left[ \sum_{\tau \in \mathcal{C}_T(\sigma_i)} f(\kappa) \cdot f(\tau) \right]$, and it represents the likelihood of observing simplex $\sigma$ given context simplices in $\mathcal{C}_T(\sigma)$. This leads to the maximization of the function:

$$\max_f \ \sum_{i=1}^{|\mathcal{W}|} \left[ -\log Z_{\sigma_i} + \sum_{\tau \in \mathcal{C}_T(\sigma_i)} f(\sigma_i) \cdot f(\tau) \right] \tag{2}$$

Our method of choice –SIMPLEX2VEC [20]– is implemented by sampling random walks from $\mathcal{H}(\mathcal{K})$ and learning simplicial embeddings with continuous-bag-of-words (CBOW) model [42]. To overcome

---

[1]Here we used $2^\mathcal{V}$ to identify the power set of the vertices.

the expensive computation of $Z_{\sigma_i}$, we train CBOW with negative sampling. While SIMPLEX2VEC is conceptually similar to $k$-SIMPLEX2VEC [19], there are important differences: (i) by fixing $k$ as simplex dimension, $k$-SIMPLEX2VEC uses exclusively upper connections through $(k+1)$-cofaces and lower connections through $(k-1)$-faces to compute random walk transitions; (ii) random walks focus on a fixed dimension, allowing the embedding computation only for $k$-simplices. SIMPLEX2VEC instead computes embedding representations for *all* simplex orders simultaneously because the random walks are sampled from the entire Hasse Diagram.

## 4  Experimental Setup

Here we describe the experimental setup used to quantify the accuracy of SIMPLEX2VEC in reconstructing and predicting higher-order interactions. In the next paragraphs, we illustrate which datasets we use, how we sample non-existing hyperlinks, and how we use them in downstream tasks.

**Table 1:** Summary statistics of empirical datasets, referring to the largest connected component of the projected graph. In order: total number of time-stamped simplices $|\mathcal{D}|$; number of unique simplices $|\mathcal{F}|$; number of training nodes $|\mathcal{V}|$ and edges $|\mathcal{E}|$ in the first 80% of $\mathcal{D}$; number of triangles in the first 80% $|\Delta|$ / new triangles in the last 20% $|\Delta^*|$; number of training tetrahedra in the first 80% $|\Theta|$ / new tetrahedra in the last 20% $|\Theta^*|$.

| Dataset | $|\mathcal{D}|$ | $|\mathcal{F}|$ | $|\mathcal{V}|$ | $|\mathcal{E}|$ | $|\Delta|/|\Delta^*|$ | $|\Theta|/|\Theta^*|$ |
|---|---|---|---|---|---|---|
| contact-high-school | 172,035 | 7,818 | 327 | 5,225 | 2,050 / 320 | 218 / 20 |
| contact-primary-school | 106,879 | 12,704 | 242 | 7,575 | 4,259 / 880 | 310 / 71 |
| email-Eu | 234,559 | 25,008 | 952 | 26,582 | 143,280 / 17,325 | 631,590 / 82,945 |
| email-Enron | 10,883 | 1,512 | 140 | 1,607 | 5,517 / 1,061 | 14,902 / 3,547 |
| tags-math-sx | 819,546 | 150,346 | 893 | 60,258 | 167,306 / 34,801 | 101,649 / 26,344 |
| congress-bills | 103,758 | 18,626 | 97 | 3,207 | 32,692 / 371 | 90,316 / 3,309 |
| coauth-MAG-History | 114,447 | 11,072 | 4,034 | 9,255 | 4,714 / 1,297 | 3,966 / 1,008 |
| coauth-MAG-Geology | 275,565 | 29,414 | 3,835 | 27,950 | 17,946 / 3,852 | 12,072 / 3,168 |

### 4.1  Data Processing

We consider data in the form of collections $\mathcal{D}$ of time-stamped interactions $\{(s_i, t_i), s_i \in \mathcal{F}, t_i \in \mathcal{T}\}_{i=1\ldots N}$, where each $s_i = (i_0, i_1, \ldots, i_k)$ is a $k$-simplex of the node set $\mathcal{V}$, $\mathcal{F}$ is the set of distinct simplices and $\mathcal{T}$ is the set of time-stamps at which interactions occur. We split $\mathcal{D}$ in two subsets, $\mathcal{D}^{train}$ and $\mathcal{D}^{test}$, corresponding to the 80th percentile $t^{(80)}$ of time-stamps, namely $\mathcal{D}^{train} = \{(s_i, t_i) \in \mathcal{D}, t^{(0)} \le t_i \le t^{(80)}\}$ and $\mathcal{D}^{test} = \{(s_i, t_i) \in \mathcal{D}, t^{(80)} < t_i \le t^{(100)}\}$, where $t^{(0)}$ and $t^{(100)}$ are the 0th and the 100th percentiles of the set $\mathcal{T}$.

We use real-world time-stamped data, indicated above with the collection $\mathcal{D}$, from different domains [9]: face-to-face proximity (`contact-high-school` and `contact-primary-school`), email exchange (`email-Eu` and `email-Enron`), online tags (`tags-math-sx`), US congress bills (`congress-bills`), coauthorships (`coauth-MAG-History` and `coauth-MAG-Geology`). When the datasets came in pairwise format, we associated simplices to cliques obtained by integrating edge information over short time intervals [9].

We considered, for all datasets, only nodes in the largest connected component of the projected graph (two nodes of the projected graph are connected if they appear in at least one simplex of $\mathcal{D}$). In addition, to lighten the embedding computations, for `congress`, `tags` and `coauth` datasets we apply a filtering approach in order to reduce their sizes: similarly to [43] with the Core set, here we selected the nodes incident in at least 5 cliques in every temporal quartile (except in `coauth-MAG-History` where we applied a threshold of 1 clique per temporal quartile). In Table 1, we report statistics for every considered dataset after the pre-processing steps (extraction of the largest projected component and filtering of unfrequent nodes).

### 4.2  Random Walk Sampling and Feature Learning

We build from $\mathcal{D}^{train}$, disregarding time-stamps, a simplicial complex $\mathcal{K}_{\mathcal{D}}^{train}$ from which we sample random walk realizations for learning low-dimensional embeddings. We consider several weighting schemes [20] to bias the random walks between the vertices $\{v_\tau\}$ of the HD:

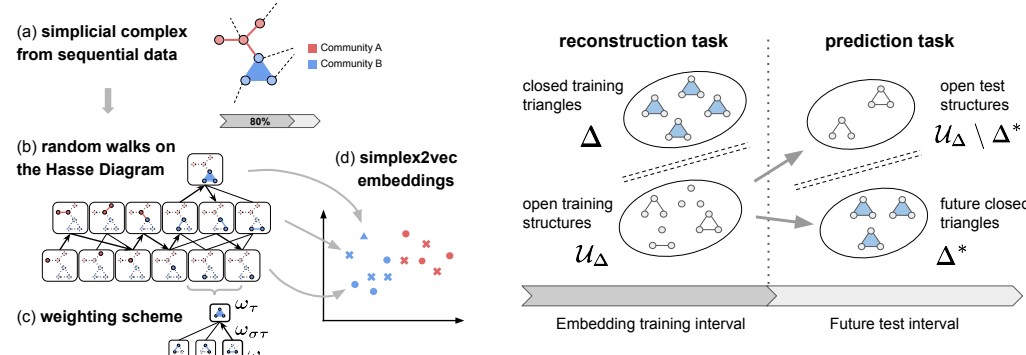

**Figure 1:** (Left) Schematic view of SIMPLEX2VEC: starting from simplicial sequential data (a), we construct a simplicial complex on whose Hasse Diagram we sample random walks (b) with different weighting (c), from which we construct the embedding space (d). (Right) Schematic description of classification tasks (reconstruction and prediction) in the case of 3-node group interactions.

- **Unweighted** The jump to a given $v_\tau$ is made by a uniform sampling among the set of neighbors $\mathcal{N}_\sigma = \mathcal{N}_\sigma^\downarrow \cup \mathcal{N}_\sigma^\uparrow$ of the node $v_\sigma$ in the HD (i.e., the sets of $(k-1)$-faces $\mathcal{N}_\sigma^\downarrow$ and $(k+1)$-cofaces $\mathcal{N}_\sigma^\uparrow$ of the $k$-simplex $\sigma$ in the simplicial complex).

- **Counts**. To every node $v_\tau$ of the HD is attached an empirical weight $\omega_\tau$, counting the number of times that $\tau$ appears in the data $\mathcal{D}$. The probability to jump from $\sigma$ to $\tau$ is given by $p_{\sigma\tau} = \frac{\omega_\tau}{\sum_{r \in \mathcal{N}_\sigma} \omega_r}$.

- **LObias**. With the definition of transition probability as before, the weight $\omega_\tau$ is defined to introduce a bias for the random walker towards low-order simplices: as explained in [20], every time a $n$-simplex $\sigma$ appears in the data its weight is increased by 1, and the weight of any subface of dimension $n - k$ is increased by $\frac{(n+1)!}{(n-k+1)!}$. There is an equivalent scheme for biasing towards high-order simplices, but we empirically observed that the performance of the first one is better.

- **EQbias**. Starting from the weight set $\{\omega_\sigma\}$ computed with empirical counts, we attach additional weights $\{\omega_{\sigma\tau}\}$ to the Hasse diagram's edges in order to have an equal probability of choosing neighbors from $\mathcal{N}_\sigma^\downarrow$ or $\mathcal{N}_\sigma^\uparrow$. Transition weights for the downward (upward) step $(\sigma, \tau)$ are defined by normalizing $\omega_\tau$ respect to all the downward (upward) weights $\omega_{\sigma\tau} \propto \frac{\omega_\tau}{\sum_{r \in \mathcal{N}_\sigma^{\downarrow(\uparrow)}} \omega_r}$, with the probability of the step given by $p_{\sigma\tau} = \frac{\omega_{\sigma\tau}}{\sum_{r \in \mathcal{N}_\sigma^\downarrow \cup \mathcal{N}_\sigma^\uparrow} \omega_{\sigma r}}$

In all experiments, we train SIMPLEX2VEC[2] on the Hasse Diagram $\mathcal{H}(\mathcal{K}_\mathcal{D}^{train})$ to obtain $d$-dimensional feature representations $\mathbf{v}_\sigma \in \mathbb{R}^d$ of every simplex $\sigma \in \mathcal{K}_\mathcal{D}^{train}$. Due to the combinatorial explosion of the number of simplicial vertices in the HD, we constrain the maximum order of the interactions to $M \in \{1, 2, 3\}$ in a reduced Hasse diagram $\mathcal{H}_M(\mathcal{K}_\mathcal{D}^{train})$ referred simply as $\mathcal{H}_M$. Consequently, every simplex with a dimension larger than $m = \max M$ is represented in $\mathcal{H}_M$ by node combinations of size up to $m$. In Fig. 1 (Left), we show the feature learning process explained before.

### 4.3 Similarity Scores and Baseline Metrics

Using the learned simplicial embeddings we assign to each higher-order link candidate $\delta$ a likelihood score based on the average pairwise inner product among 0-simplex embeddings of nodes $\{\mathbf{v}_i, i \in \delta\}$ or any high-order $k$-simplices $\{\mathbf{v}_\sigma, \sigma \subset \delta\}$:

$$s_k(\delta) = \frac{1}{|\binom{\binom{\delta}{k+1}}{2}|} \sum_{(\sigma,\tau) \in \binom{\binom{\delta}{k+1}}{2}} \mathbf{v}_\sigma \cdot \mathbf{v}_\tau \tag{3}$$

---

[2]We used the WORD2VEC implementation from Gensim (https://radimrehurek.com/gensim/) and ran the CBOW model with window $T = 10$ and 5 epochs. We sample 10 random walks of length 80 per simplex as input to WORD2VEC.

To assess the reconstruction and prediction performances of the embedding model, we compare likelihood scores defined in Eq. 3 with other baseline metrics:

- **Projected metrics**. Local and global node-level features computed from the projected graph. The projected graph is defined as $\mathcal{G}_{\mathcal{D}}^{train} = (\mathcal{V}, \mathcal{E})$, where $\mathcal{V}$ is the set of 0-simplices of the complex $\mathcal{K}_{\mathcal{D}}^{train}$ and $\mathcal{E} = \left\{ s \in \mathcal{K}_{\mathcal{D}}^{train} : |s| = 2 \right\}$ is the set of links between training nodes that interacted in at least one simplex of $\mathcal{D}^{train}$. Moreover, edges $(i, j)$ can be weighted with the number of simplices of $\mathcal{D}$ containing both $i$ and $j$. For triangles-related tasks we considered several 3-way metrics computed with the code[3] released by [9] (we show the best performant: *Harmonic mean*, *Geometric mean*, *Katz*, *PPR*, *Logistic Regression*). We exploited also the pair-wise random walk measure PPMI$_T$ [44], for tetrahedra-related tasks where 4-way implementations of the above listed scores are not available. PPMI is widely used as a similarity function for node embeddings, and variations of the window size $T$ allow us to take into account both local and global information.

- **Spectral embedding**. Features from the spectral decomposition of the combinatorial $k$-Laplacian [45]. Given the set of *boundary* matrices $\{\mathbf{B}_k\}$, which incorporate incidence relationships between $k$-simplices and their $(k-1)$-faces[4], the unweighted $k$-Laplacian is $\mathbf{L}_k = \mathbf{B}_k^{\mathrm{T}}\mathbf{B}_k + \mathbf{B}_{k+1}\mathbf{B}_{k+1}^{\mathrm{T}}$. We consider also the weighted $k$-Laplacian [46], calculated with the substitutions $\mathbf{B}_k \rightarrow \mathbf{W}_{k-1}^{-1/2}\mathbf{B}_k\mathbf{W}_k^{1/2}$, where every $\mathbf{W}_k$ is a diagonal matrix containing empirical counts of any $k$-simplex[5]. Following the same procedure used in graph spectral embeddings [47], we compute the eigenvectors matrix $\mathbf{Q}_k \in \mathbb{R}^{n_k \times d}$ corresponding to the first $d$ smallest non-zero eigenvalues of $\mathbf{L}_k$ and we use the rows of $\mathbf{Q}_k$ as $d$-dimensional spectral embeddings for $k$-simplices.

- **k-SIMPLEX2VEC embedding**. Features learned with an extension of NODE2VEC [19] that samples random walks from higher-order transition probabilities[6] (e.g., edge-to-edge occurrences) in a single simplicial dimension. This model is based on sampling from a uniform structure without taking into account simplicial weights.

Likelihood scores of candidate higher-order links are assigned for the embedding models with the same metric of Eq. 3 used for SIMPLEX2VEC embeddings. In $k$-SIMPLEX2VEC, we sample the same number of random walks per simplex, with the same length, as the ones used for SIMPLEX2VEC.

## 4.4 Downstream Tasks and Open Configurations Sampling

Similarly to the standard graph case, non-existing links are usually the majority class and this imbalance is even more pronounced in the higher-order case [30] (in graphs we have $\mathcal{O}(|\mathcal{V}|^2)$ potential links, but the number of potential hyperlinks/simplices is $\mathcal{O}(2^{|\mathcal{V}|})$ in higher-order structures). To compensate, we focus the work on 3-node and 4-node groups, reducing the number of potential hyperedges to $\mathcal{O}(|\mathcal{V}|^3)$ and $\mathcal{O}(|\mathcal{V}|^4)$ respectively. For a concise presentation, in the next paragraphs we describe mainly the 3-way case. Hence, we restrict the set of possible interactions $\mathcal{S}$ to be exclusively closed triangles $\Delta$ of the training complex and the corresponding 3-node complementary set $\mathcal{U}_\Delta$:

$$\Delta = \left\{ s \in \mathcal{K}_{\mathcal{D}}^{train} : |s| = 3 \right\}, \quad \mathcal{U}_\Delta = \binom{\mathcal{V}}{3} \setminus \Delta \tag{4}$$

where we used $\binom{\mathcal{V}}{3}$ as the set of 3-node combinations of elements from $\mathcal{V}$ (we instead denote respectively with $\Theta$ and $\mathcal{U}_\Theta$ the observed and unobserved tetrahedra of the training set). With the reconstruction task we aim to discern those triplets $\delta$ interacting as a 2-simplex in the training window $[0, t^{(80)}]$, and so $\delta \in \Delta$, from those that are groups of lower-order simplices, meaning $\delta \in \mathcal{U}_\Delta$. Moreover, defining $\Delta^*$ as the set of new triadic interactions in the interval $(t^{(80)}, t^{(100)}]$, with the prediction task we aim to classify those open groups $\bar{\delta} \in \mathcal{U}_\Delta$ that give rise to a simplicial closure on the test time-span ($\bar{\delta} \in \Delta^*$) respect to those ones that remain open ($\bar{\delta} \in \mathcal{U}_\Delta \setminus \Delta^*$). In Figure 1 (Right), we sketch the task's formulation based on 2-simplices (3-node configurations).

---

[3] `https://github.com/arbenson/ScHoLP-Tutorial`

[4] Boundary matrix $\mathbf{B}_k \in \{0, \pm 1\}^{n_{k-1} \times n_k}$ requires the definition of oriented simplices, see [2] for additional details.

[5] Weights matrices satisfy the consistency relations $\mathbf{W}_k = |\mathbf{B}_{k+1}|\mathbf{W}_{k+1}$, see [46] for further details.

[6] `https://github.com/celiahacker/k-simplex2vec`

**Table 2:** Number of unobserved configurations obtained with the sampling approach in different datasets.

| Dataset | Unseen configurations sampled from $\mathcal{U}_\Delta$ $n_\mathcal{E}(\times 10^3)$ | | | |
|---|---|---|---|---|
| | 0 | 1 | 2 | 3 |
| `contact-high-school` | 3,476 | 1,150 | 107 | 25 |
| `email-Eu` | 8,096 | 1,392 | 1,654 | 186 |
| `tags-math-sx` | 6,229 | 2,473 | 5,467 | 1,725 |
| `coauth-MAG-History` | 9,958 | 30 | 60 | 2 |

| Dataset | Unseen configurations sampled from $\mathcal{U}_\Theta$ $n_\Delta(\times 10^3)$ | | | | |
|---|---|---|---|---|---|
| | 0 | 1 | 2 | 3 | 4 |
| `contact-primary-school` | 17,683 | 396 | 19 | 2 | < 1 |
| `email-Enron` | 7,048 | 400 | 28 | 2 | < 1 |
| `congress-bills` | 1,462 | 1,264 | 325 | 149 | 80 |
| `coauth-MAG-Geology` | 15,473 | 593 | 30 | 3 | < 1 |

To overcome the impossibility of enumerating all the unseen configurations, we collect negative instances for the classification tasks by sampling fixed-size groups of nodes. In practice, we sample *stars*, *cliques* and other network *motifs* [39] from the projected graph to collect group configurations with distinct densities of lower-order interactions. We independently sample nodes to obtain (more likely) groups with unconnected units. For each sampled 3-node group $\delta$ we count the number of involved training edges $n_\mathcal{E}(\delta)$, and we analyze tasks performances for open configurations characterized by fixing $n_\mathcal{E}(\delta) \in \{0, 1, 2, 3\}$. For 4-node configurations, instead of $n_\mathcal{E}(\delta)$, we consider the number of training triangles $n_\Delta(\delta) \in \{0, 1, 2, 3, 4\}$ to differentiate open groups. In Table 2 we report the number of open configurations randomly selected from $\mathcal{U}_\Delta$ and $\mathcal{U}_\Theta$. We extracted with replacement $10^7$ samples of candidate open configurations for each pattern (*stars*, *cliques*, *motifs*, and *independent* node groups).

We claim that quantities $n_\mathcal{E}(\delta)$ and $n_\Delta(\delta)$ are related to the concept of $hardness$ of non-hyperlinks [39], i.e. the propensity of open groups to be misclassified as closed interaction, and they influence the difficulty of downstream classification tasks. In fact, increasing the number of lower-order faces -$n_\mathcal{E}$ or $n_\Delta$- engaged into a fake hyperlink, the latter becomes more and more structurally similar to true simplices, making the classification task more difficult.

## 5    Results and Discussion

With the previously described setup, we conducted experiments with 3-node configurations on datasets `contact-high-school`, `email-Eu`, `tags-math-sx`, `coauth-MAG-History` and with 4-node configurations on the remaining ones. Due to the limited space available, we only report 3-way results leaving the 4-way analysis in the Appendix. We also include supplemental experiments with hypergraph-based embeddings not shown in the main text.

We highlight the classification performance when using different embedding similarities $s_k(\delta)$ on open configurations with different $n_\mathcal{E}(\delta)$ (in the case of triangles, or $n_\Delta(\delta)$ for tetrahedra). For each case, triangles and tetrahedra classification, we examine: (i) the comparison with $k$-SIMPLEX2VEC embeddings in the *unweighted* scenario, to study how different embedding models learn statistical patterns from the simplicial structure; (ii) the comparison with classical metrics in the *weighted* scenario, to study how the addition of empirical weights influences the embedding performance with respect to traditional weighted approaches.

Results are presented in terms of average binary classification scores, where test sets are generated by randomly chosen open and closed groups. Contrarily to previous work [9, 35], we evaluate models without a fixed class imbalance because we cannot access the entire negative classes (e.g., $\mathcal{U}_\Delta$ and $\mathcal{U}_\Delta \setminus \Delta^*$ respectively in 3-way reconstruction and prediction). Instead, in every test set we uniformly sample the cardinality of the two classes to be between 1 and the number of available samples according to the task. We report calibrated AUC-PR scores [48] to account for the difference in class

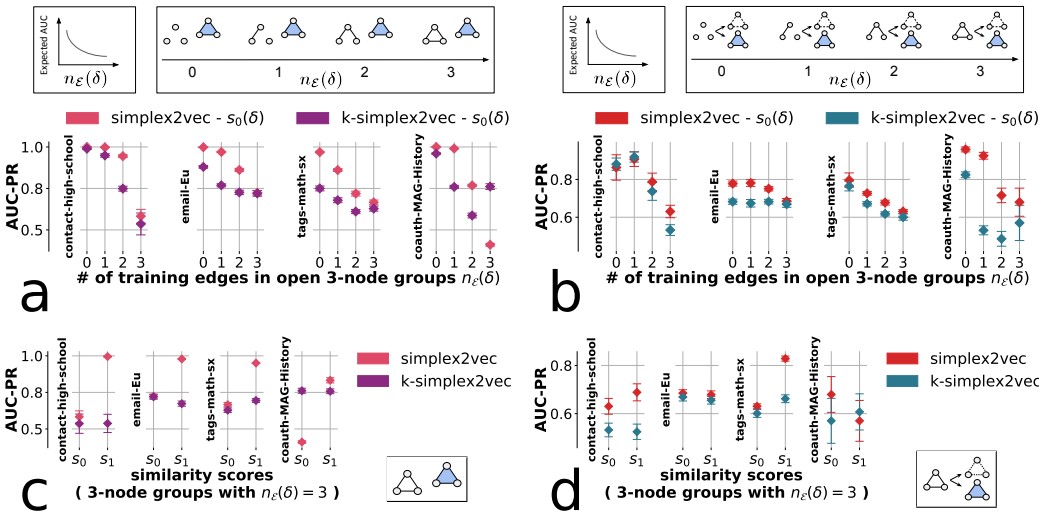

**Figure 2:** Calibrated AUC-PR scores on 3-way link reconstruction (a)(c) and prediction (b)(d) for SIMPLEX2VEC and $k$-SIMPLEX2VEC with: (a)(b) similarity $s_0$ varying the parameter $n_\mathcal{E}$; (c)(d) similarity $s_k$ (with $k$ in $\{0, 1\}$) on highly edge-dense open configurations ($n_\mathcal{E} = 3$). Metrics are computed in unweighted representations, with SIMPLEX2VEC trained on $\mathcal{H}_{k+1}$ when showing results for metric $s_k$. The label unbalancing in each sample is uniformly drawn between 1:1 and 1:5000. A schematic view of positive and negative examples is reported for each classification task.

imbalance as a consequence of our sampling choice[7]. In Figure 2, for a fair comparison with the other projected and embedding metrics, we report the similarity $s_k$ training SIMPLEX2VEC on $\mathcal{H}_{k+1}$. For instance, when comparing node embedding performance ($k$=0), we use the Hasse Diagram $\mathcal{H}_1$ to neglect triadic and higher-order information not explicitly incorporated with node-to-node proximities in $k$-SIMPLEX2VEC and spectral node embeddings. Best average scores are chosen for embedding models with a search on vector sizes in the set $\{8, 16, 32, 64, 128, 256, 512, 1024\}$.

## 5.1 Reconstruction and Prediction of 3-way Interactions: the Unweighted Scenario and $k$-SIMPLEX2VEC

### 5.1.1 Comparison of Pair-wise Node Proximities

In Figure 2(a)(b), we show evaluation metrics on higher-order link classification (reconstruction and prediction) for 3-way interactions, computed with *unweighted* node-level information from different models, varying the quantity $n_\mathcal{E}(\delta)$ referred to the open configurations. We recall that in this case $k$-SIMPLEX2VEC is equivalent to the NODE2VEC embedding of the projected graph. Hasse diagram $\mathcal{H}_1$ scores $s_0(\delta)$ computed with SIMPLEX2VEC perform overall better than proximities of the projected graph (i.e., $k$-SIMPLEX2VEC scores) in almost all cases, meaning that the information given by the pairwise structures is enriched by considering multiple layers of interactions, even without leveraging interaction weights (both in $\mathcal{G}_\mathcal{D}^{train}$ and $\mathcal{K}_\mathcal{D}^{train}$).

Generally, we observe an expected decrease in performance for every model with respect to parameter $n_\mathcal{E}$. For example, a few datasets show less sensitivity in the performance of prediction tasks to variations of $n_\mathcal{E}(\delta)$ (e.g., email-Eu). We ascribe this difference to domain-specific effects and peculiarities of those datasets. Embedding similarity $s_0(\delta)$ from $\mathcal{H}_1$ diagram outperforms $k$-SIMPLEX2VEC proximities in almost every reconstruction task, except for coauth-MAG-History on open configurations with $n_\mathcal{E} = 3$. This fact seems connected with some specific graph features of collaborations (even possibly related to the filtering approach utilized). Moreover, coauthorship relations usually are not characterized by subset dependencies (writing a paper as a group does not imply pair-wise collaborations [3]) that are encoded with simplicial complexes. In prediction tasks, we observe the

---

[7]For this purpose we fix the reference class ratio $\pi_0 = 0.5$. See [48] for additional details. We also tested the AUC-ROC metric with similar findings.

**Table 3:** Calibrated AUC-PR scores for higher-order link reconstruction (Top) and prediction (Bottom) on 3-node groups, with the hardest class of negative configurations ($n_{\mathcal{E}} = 3$). The best scores for different methods are reported in boldface letters; among these ones, the best overall score is blue-shaded and the second best score is grey-shaded.

| Features Type | | | Dataset | | | | | | | |
|---|---|---|---|---|---|---|---|---|---|---|
| | | | contact-high-school | | email-Eu | | tags-math-sx | | coauth-MAG-History | |
| | | | $s_0(\delta)$ | $s_1(\delta)$ | $s_0(\delta)$ | $s_1(\delta)$ | $s_0(\delta)$ | $s_1(\delta)$ | $s_0(\delta)$ | $s_1(\delta)$ |
| Neural Embedding | Hasse diagram $\mathcal{H}_1$ | Unweighted | 57.5±1.9 | 51.4±1.2 | 72.0±0.3 | 64.0±0.2 | 66.7±0.2 | 57.1±0.1 | 41.1±0.9 | 75.5±1.1 |
| | | Counts | 79.5±1.0 | 84.4±0.9 | 76.3±0.4 | 73.3±0.2 | 80.5±0.1 | **87.8±0.1** | 41.6±1.0 | 76.0±1.1 |
| | | LObias | 81.6±2.4 | **89.5±0.8** | 76.1±0.3 | 71.2±0.2 | 76.9±0.1 | 83.7±0.1 | 41.7±0.7 | 57.7±1.2 |
| | Hasse diagram $\mathcal{H}_2$ | Unweighted | 55.5±3.0 | **99.5±0.1** | 61.0±0.4 | **97.9±0.0** | 66.7±0.1 | **95.1±0.0** | 40.0±0.5 | 83.1±1.3 |
| | | Counts | 57.0±1.3 | 91.2±0.9 | 54.5±0.2 | 92.6±0.1 | 66.2±0.1 | 89.4±0.1 | 35.3±0.4 | 82.1±1.3 |
| | | LObias | 84.7±2.2 | 91.9±0.8 | 80.6±0.3 | 81.6±0.2 | 77.9±0.1 | 84.3±0.1 | 57.3±1.0 | 70.4±1.4 |
| | | EQbias | 72.7±1.1 | 89.2±0.7 | 71.8±0.3 | 75.0±0.2 | 78.2±0.2 | 88.0±0.1 | 39.3±0.7 | **87.3±1.1** |
| Spectral Embedding | Combinatorial Laplacians | Unweighted | 52.4±3.7 | **77.0±1.3** | 67.3±0.3 | 65.3±0.2 | 58.4±0.2 | 50.7±0.1 | 72.1±1.1 | 63.5±1.4 |
| | | Weighted | 70.4±1.6 | 75.3±1.6 | **79.4±0.2** | 76.4±0.1 | **79.9±0.1** | 50.4±0.1 | **82.3±1.0** | 68.4±1.2 |
| Projected Metrics | Harm. mean | Weighted | 85.5±1.5 | | **74.0±0.2** | | 83.1±0.1 | | 53.3±1.1 | |
| | Geom. mean | | **85.8±1.1** | | 72.5±0.2 | | **86.8±0.1** | | 52.9±1.3 | |
| | Katz | | 78.6±1.1 | | 65.6±0.2 | | 81.8±0.1 | | 49.2±1.5 | |
| | PPR | | 76.9±1.4 | | 70.7±0.2 | | 81.8±0.1 | | **74.8±1.3** | |

| Features Type | | | Dataset | | | | | | | |
|---|---|---|---|---|---|---|---|---|---|---|
| | | | contact-high-school | | email-Eu | | tags-math-sx | | coauth-MAG-History | |
| | | | $s_0(\delta)$ | $s_1(\delta)$ | $s_0(\delta)$ | $s_1(\delta)$ | $s_0(\delta)$ | $s_1(\delta)$ | $s_0(\delta)$ | $s_1(\delta)$ |
| Neural Embedding | Hasse diagram $\mathcal{H}_1$ | Unweighted | 62.9±5.2 | 50.6±4.7 | 68.5±0.7 | 57.6±0.5 | 63.2±0.3 | 54.0±0.5 | **69.5±8.2** | 63.2±6.6 |
| | | Counts | **74.2±3.0** | 73.0±3.4 | **74.3±0.8** | 67.3±0.7 | 74.3±0.4 | **84.0±0.3** | 68.7±8.4 | 66.6±8.6 |
| | | LObias | 70.6±2.8 | 65.6±5.3 | 70.5±0.6 | 64.5±0.8 | 71.3±0.5 | 79.1±0.5 | 68.8±8.7 | 66.5±8.7 |
| | Hasse diagram $\mathcal{H}_2$ | Unweighted | 62.5±6.3 | 69.5±4.9 | 66.2±0.7 | 67.8±0.6 | 62.5±0.2 | **83.1±0.2** | 65.9±8.5 | 55.6±8.0 |
| | | Counts | 64.3±3.6 | 72.8±3.6 | 61.8±0.7 | 69.1±0.6 | 62.9±0.3 | 82.3±0.3 | 67.3±8.2 | 61.0±9.6 |
| | | LObias | 69.7±3.5 | 65.4±5.1 | 69.0±0.6 | 60.3±0.6 | 71.2±0.7 | 79.2±0.4 | 67.3±7.9 | 64.2±9.6 |
| | | EQbias | 72.4±3.6 | **73.5±3.5** | **71.3±0.6** | 66.1±0.6 | 71.2±0.4 | 82.3±0.3 | **67.8±8.6** | 65.7±9.3 |
| Spectral Embedding | Combinatorial Laplacians | Unweighted | 56.4±3.6 | 56.7±6.8 | 63.8±0.6 | 53.5±0.7 | 55.1±0.2 | 50.4±0.2 | 57.8±6.0 | 56.4±5.7 |
| | | Weighted | **66.5±5.3** | 56.1±6.5 | 65.2±0.8 | 55.6±0.7 | **72.8±0.4** | 50.3±0.3 | **70.1±8.3** | 53.5±6.8 |
| Projected Metrics | Harm. mean | Weighted | 71.4±4.3 | | 64.5±0.8 | | 79.0±0.2 | | 61.6±8.2 | |
| | Geom. mean | | **73.1±3.8** | | 66.7±0.8 | | **83.3±0.2** | | 62.4±7.7 | |
| | Katz | | 69.3±3.7 | | 63.2±0.6 | | 77.8±0.3 | | 62.4±7.0 | |
| | PPR | | 69.8±3.9 | | **68.8±0.5** | | 75.7±0.4 | | 57.7±4.6 | |
| | Logistic Regression | Unweighted | 68.7±3.1 | | 68.1±0.7 | | 81.2±0.2 | | **65.4±6.9** | |

same advantage of SIMPLEX2VEC respect to $k$-SIMPLEX2VEC, except in `contact-high-school` where the models perform similarly on $n_{\mathcal{E}} < 2$.

### 5.1.2 Comparison of Higher-order Edge Proximities

In the previous sections, the metric $s_0(\delta)$ was computed from feature representations of 0-simplices. Here we analyze instead how performances change when we use embedding representations of 1-simplices (edge representations) to compute $s_1(\delta)$. Intuitively, group representations like 1-simplex embeddings should convey higher-order information useful to improve classification with respect to node-level features.

In Figure 2(c)(d), we show evaluation metrics on higher-order link classification for 3-way interactions, comparing *unweighted* node-level and edge-level information from different models, fixing the quantity $n_{\mathcal{E}}(\delta) = 3$ referred to the open configurations. We consider fully connected triangle configurations because, besides being the harder configurations to be classified, they consist of the set of links necessary to compute $s_1(\delta)$.

Generally, we notice an increase in classification scores when using $s_1(\delta)$ similarity rather $s_0(\delta)$ with SIMPLEX2VEC embeddings, instead $k$-SIMPLEX2VEC exhibits reduced gains in most datasets. The SIMPLEX2VEC performance gain is quite large (between 30% and 100%) in all reconstruction tasks, and for prediction tasks it is noticeable on `contact-high-school` and `tags-math-sx` while it is even negative on `coauth-MAG-History`. Regarding the latter dataset, the use of edge-level similarity balances the node-level reconstruction loss noticed in Figure 2(a).

## 5.2 Reconstruction and Prediction of 3-way Interactions: Role of Simplicial Weights

Previously we showed that feature representations learned through the hierarchical organization of the HD enhance the classification accuracy of closed triangles when considering unweighted complexes. We now integrate these results by studying the effect of introducing weights. In particular, we analyze the importance of weighted interactions in our framework, focusing on the case where fully connected open triangles are the negative examples for downstream tasks.

In Table 3 (Top) we show higher-order link reconstruction results: simplicial similarity $s_1(\delta)$ on the unweighted HD $\mathcal{H}_2$ outperforms all other methods, in particular weighted metrics based on Laplacian similarity and projected graph geometric mean, allowing almost perfect reconstruction in 3 out of 4 datasets. Compared with projected graph metrics, this was expected since 3-way information is incorporated in $\mathcal{H}_2$, and the optimal scores reflect the goodness of fit of the embedding algorithm. Weighting schemes *Counts* and *EQbias* also obtain excellent scores with $s_1(\delta)$ metric, while metric $s_0(\delta)$ benefits from the use of *LObias* weights. Differently, even simplicial similarity $s_1(\delta)$ on Hasse diagram $\mathcal{H}_1$ outperforms baseline scores in half of the datasets (with weighting schemes *Counts* and *LObias*), showing the feasibility of reconstructing 2-order interactions from weighted lower-order simplices (vertices in $\mathcal{H}_1$ are simplices of dimension 0 and 1) similarly to previous work on hypergraph reconstruction [8].

In Table 3 (Bottom) we show higher-order link prediction results. Overall, SIMPLEX2VEC embeddings trained on $\mathcal{H}_1$ with *Counts* and *EQbias* weights give better results: in contact-high-school and email-Eu with $s_0(\delta)$ metric, in tags-math-sx with $s_1(\delta)$ metric. In dataset coauth-MAG-History the unweighted $s_0(\delta)$ score is outperformed uniquely by the weighted $\mathbf{L}_0$ embedding, with weighted simplicial counterparts resulting in similar performances. In the space of projected graph scores, good results are obtained with *geometric mean* and *logistic regression*, which were among the best metrics in one of the seminal works on higher-order link prediction [9].

Finally, we observe that weighting schemes for neural simplicial embeddings overall positively contribute to classification tasks both for reconstruction and prediction.

# 6 Conclusions and Future Work

In this paper, we introduced SIMPLEX2VEC for representation learning on simplicial complexes. In particular, we focused on formalizing reconstruction and link prediction tasks for higher-order structures, and we tested the proposed model on solving such downstream tasks. We showed that SIMPLEX2VEC-based representations are more effective in classification than traditional approaches and previous higher-order embedding methods. In particular, we prove the feasibility of using simplicial embedding of Hasse diagrams in reconstructing system's polyadic interactions from lower-order edges, in addition to adequately predicting future simplicial closures. SIMPLEX2VEC enables the investigation of the impact of different topological features, and we showed that weighted and unweighted models have different predictive power. Future work should focus on understanding these differences through the analysis of link predictability [49,50] with higher-order edges as a function of datasets' peculiarities. Future work includes algorithmic approaches to tame the scalability limits set by the combinatorial structure of the Hasse diagram, which could for example be tackled via different optimization frameworks [51,52] and hierarchical approaches [18,53].

## Author Contributions

SP, AP and GP conceived and designed the study, performed the analysis and wrote the manuscript. All authors read and approved the final manuscript.

## Acknowledgements

The authors thank Prof. Alain Barrat and Prof. Ciro Cattuto for the valuable discussions that helped shape this research work.

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

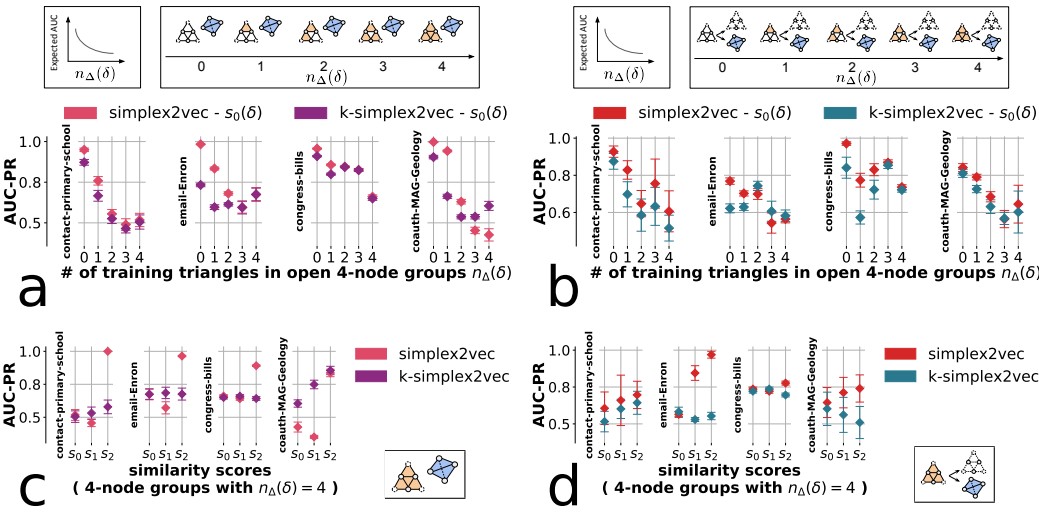

**Figure A1:** Calibrated AUC-PR scores on 4-way link reconstruction (a)(c) and prediction (b)(d) for SIMPLEX2VEC and $k$-SIMPLEX2VEC with: (a)(b) similarity $s_0$ varying the parameter $n_\Delta$; (c)(d) similarity $s_k$ (with $k$ in $\{0, 1, 2\}$) on highly triangle-dense open configurations ($n_\Delta = 4$). Metrics are computed in unweighted representations, with SIMPLEX2VEC trained on $\mathcal{H}_{k+1}$ when showing results for metric $s_k$. Label unbalancing in each sample is uniformly drawn between 1:1 and 1:5000. A schematic view of positive and negative examples is reported for each classification task.

## A    Appendix

### A.1    Beyond 3-way Interactions: Tetrahedra

*Unweighted Analysis.* In Figure A1(a), we show node-level evaluation metrics for 4-way higher-order reconstruction. Metric $s_0(\delta)$ of SIMPLEX2VEC computed on $\mathcal{H}_1$ shows overall slightly better performances respect to $k$-SIMPLEX2VEC similarities, especially when the density of triangles is low ($n_\Delta < 3$). In coauth-MAG-Geology we observe also a remarkable increment of $k$-SIMPLEX2VEC reconstruction scores for negative examples with increasing $n_\Delta(\delta)$, and this is also observable in email-Enron. In Figure A1(b), we report node-level evaluation metrics for 4-way higher-order prediction. Node-level SIMPLEX2VEC embedding performs better than $k$-SIMPLEX2VEC, on contact-primary-school and, to a lesser extent, on coauth-MAG-Geology. In email-Enron and congress-bills SIMPLEX2VEC performance increases when the density of triangles is low ($n_\Delta \leq 2$). Higher-order similarity measures from $k$-SIMPLEX2VEC, in Figure A1(c)(d), are outperformed by the SIMPLEX2VEC ones in many cases, especially $s_2(\delta)$ metric for contact-primary-school, email-Enron and congress-bills in reconstruction tasks. In prediction tasks with email-Enron and coauth-MAG-Geology SIMPLEX2VEC obtain mainly good results overcoming the simplicial baseline. These results generally confirm our previous findings on 3-way tasks, which displayed an increasing classification capability when using higher-order proximities $s_k$ ($k > 0$) for SIMPLEX2VEC.

*Weighted Analysis.* In Table A1 (Top) we show reconstruction scores of tetrahedra, when simplicial embeddings are trained on Hasse diagram $\mathcal{H}_2$ and negative examples are given by open 4-way configurations with four triangular faces. Due to $\mathcal{H}_2$ characteristics, features learned from the simplicial complex are not aware of tetrahedral structures and this task results on reconstructing 4-node groups from training data with most triadic structures. Previous work analyzed the problem of higher-order edge reconstruction from pair-wise data [8], but here we focus on a not previously studied task based on triadic data. From the comparison with spectral embeddings and PPMI proximities, we notice that SIMPLEX2VEC weighted $s_2(\delta)$ similarity (*LObias* and *EQbias*) is the best on half of the datasets in classifying closed tetrahedra respect to triangle-rich open groups. In email-Enron weighted $\mathbf{L}_1$ embedding outperforms the unweighted (and weighted ones) $s_0(\delta)$ simplicial metric, while in coauth-MAG-Geology the best score is given by the unweighted PPMI$_1$ (which is also the best projected metric in the other 3 datasets). In Table A1 (Bottom) we report classification scores for the prediction of simplicial closures on tetrahedra when neural embeddings are trained on Hasse

**Table A1:** Calibrated AUC-PR scores for higher-order link reconstruction (Top) and prediction (Bottom) on 4-node groups, with the hardest class of negative configurations ($n_\Delta = 4$). The best scores for different methods are reported in boldface letters; among these ones, the best overall score is blue-shaded and the second best score is grey-shaded.

| Dataset | | Neural Embedding (Hasse Diagram $\mathcal{H}_2$) | | | | Spectral Embedding (Combinatorial Laplacians) | | | | Projected Graph PPMI Metric | | |
|---|---|---|---|---|---|---|---|---|---|---|---|---|
| | | $s_0(\delta)$ | $s_1(\delta)$ | $s_2(\delta)$ | | $s_0(\delta)$ | $s_1(\delta)$ | $s_2(\delta)$ | | $T=1$ | $T=10$ | $T=\infty$ |
| contact-primary-school | *Unweighted* | 52.9±3.3 | 45.2±2.7 | 64.5±2.8 | *Unweighted* | 52.1±3.8 | **58.2±2.0** | 53.4±3.0 | | **51.5±3.1** | 50.2±3.0 | 50.2±3.0 |
| | *Counts* | 48.4±3.0 | 46.2±2.8 | 59.1±3.3 | | | | | | | | |
| | *LObias* | 50.6±3.2 | 61.6±3.3 | **70.7±3.9** | *Weighted* | 54.0±2.8 | 55.9±2.8 | 53.4±2.1 | | 47.9±3.1 | 47.0±2.7 | 48.5±2.5 |
| | *EQbias* | 45.2±3.6 | 47.0±3.0 | 58.5±3.3 | | | | | | | | |
| email-Enron | *Unweighted* | **69.0±0.4** | 56.0±0.4 | 58.2±0.3 | *Unweighted* | 69.0±0.5 | 68.0±0.4 | 55.5±0.3 | | **68.5±0.4** | 66.7±0.5 | 66.9±0.4 |
| | *Counts* | 60.6±0.5 | 61.3±0.5 | 54.0±0.4 | | | | | | | | |
| | *LObias* | 68.0±0.5 | 46.5±0.5 | 57.4±0.5 | *Weighted* | 71.1±0.4 | **79.0±0.3** | 76.9±0.2 | | 58.3±0.4 | 57.9±0.5 | 62.0±0.5 |
| | *EQbias* | 62.1±0.7 | 44.4±0.3 | 53.1±0.4 | | | | | | | | |
| congress-bills | *Unweighted* | 63.1±0.2 | 64.4±0.1 | 51.8±0.2 | *Unweighted* | 56.1±0.2 | 58.4±0.1 | 49.8±0.1 | | 65.9±0.1 | **66.0±0.1** | 65.9±0.1 |
| | *Counts* | 43.1±0.1 | 70.4±0.1 | 72.5±0.1 | | | | | | | | |
| | *LObias* | 49.0±0.1 | **74.2±0.1** | 60.6±0.2 | *Weighted* | 55.0±0.1 | **62.8±0.2** | 55.3±0.2 | | 49.1±0.1 | 47.8±0.1 | 47.3±0.1 |
| | *EQbias* | 65.7±0.2 | 69.0±0.1 | **74.2±0.1** | | | | | | | | |
| coauth-MAG-Geology | *Unweighted* | 71.6±0.5 | 34.6±0.3 | **84.2±0.7** | *Unweighted* | 62.6±0.6 | 61.7±0.9 | 49.3±0.9 | | **86.0±0.4** | 77.8±0.4 | 75.5±0.5 |
| | *Counts* | 40.5±0.3 | 36.2±0.4 | 74.1±0.3 | | | | | | | | |
| | *LObias* | 64.1±0.5 | 34.4±0.3 | 73.3±0.5 | *Weighted* | **85.8±0.7** | 65.7±0.5 | 44.9±0.7 | | 76.3±0.6 | 71.9±0.5 | 70.6±0.6 |
| | *EQbias* | 36.7±0.3 | 37.5±0.2 | 79.2±0.4 | | | | | | | | |

| Dataset | | Neural Embedding (Hasse Diagram $\mathcal{H}_3$) | | | | Spectral Embedding (Combinatorial Laplacians) | | | | Projected Graph PPMI Metric | | |
|---|---|---|---|---|---|---|---|---|---|---|---|---|
| | | $s_0(\delta)$ | $s_1(\delta)$ | $s_2(\delta)$ | | $s_0(\delta)$ | $s_1(\delta)$ | $s_2(\delta)$ | | $T=1$ | $T=10$ | $T=\infty$ |
| contact-primary-school | *Unweighted* | 56.4±1.8 | 58.6±2.3 | 66.8±2.4 | *Unweighted* | 82.1±4.0 | 85.4±1.7 | **85.9±3.1** | | 49.3±2.2 | 45.8±1.6 | 45.7±1.7 |
| | *Counts* | 63.0±2.7 | 67.8±0.7 | **72.2±1.6** | | | | | | | | |
| | *LObias* | 60.4±1.6 | 61.2±2.2 | 62.4±2.6 | *Weighted* | 57.8±2.4 | 81.3±4.4 | 70.6±1.5 | | **61.1±2.3** | 47.4±1.6 | 48.6±1.6 |
| | *EQbias* | 62.7±2.0 | 65.6±1.2 | 68.3±2.2 | | | | | | | | |
| email-Enron | *Unweighted* | 88.3±6.6 | **98.0±2.1** | 96.9±2.3 | *Unweighted* | 92.7±2.9 | 67.6±5.7 | **97.1±1.8** | | 50.3±0.2 | 50.9±0.5 | 50.8±0.5 |
| | *Counts* | 77.0±5.6 | 88.7±4.0 | 83.5±4.5 | | | | | | | | |
| | *LObias* | 60.5±3.1 | 73.7±5.4 | 88.4±4.0 | *Weighted* | 84.8±5.6 | 88.7±3.7 | 95.8±2.4 | | **55.8±2.2** | 53.3±1.3 | 54.7±1.5 |
| | *EQbias* | 57.9±2.5 | 84.9±3.6 | 80.4±5.6 | | | | | | | | |
| congress-bills | *Unweighted* | 47.9±0.1 | 34.0±0.0 | **77.7±0.3** | *Unweighted* | 60.8±0.2 | **64.3±0.3** | 48.8±0.2 | | **74.7±0.2** | **74.7±0.2** | **74.7±0.2** |
| | *Counts* | 49.9±0.2 | 37.4±0.1 | 74.6±0.3 | | | | | | | | |
| | *LObias* | 40.2±0.2 | 76.9±0.3 | 74.0±0.3 | *Weighted* | 40.2±0.1 | 53.1±0.3 | 50.8±0.2 | | 40.2±0.1 | 40.8±0.1 | 40.2±0.1 |
| | *EQbias* | 64.2±0.2 | 58.4±0.3 | 71.4±0.2 | | | | | | | | |
| coauth-MAG-Geology | *Unweighted* | 55.1±7.7 | 60.1±7.2 | 74.8±4.8 | *Unweighted* | 57.0±6.9 | 48.1±7.8 | 52.1±7.3 | | 50.7±3.5 | 54.6±6.3 | 55.3±7.4 |
| | *Counts* | 54.0±5.9 | 74.1±3.6 | 78.6±4.4 | | | | | | | | |
| | *LObias* | 75.9±5.0 | **84.2±2.9** | 73.9±4.3 | *Weighted* | **88.5±3.2** | 52.0±7.7 | 52.7±7.3 | | 54.9±4.5 | **56.1±5.9** | 55.3±4.8 |
| | *EQbias* | 51.3±4.7 | 76.1±4.3 | 72.8±6.1 | | | | | | | | |

diagram $\mathcal{H}_3$ (we empirically observed better results with respect to $\mathcal{H}_2$). We compare these results with spectral embeddings and PPMI projected metrics in predicting which mostly triangle-dense configurations will close in a tetrahedron in the last 20% of data. Unusually, best scores obtained with SIMPLEX2VEC come from the unweighted setting in email-Enron and congress-bills with respectively $s_1(\delta)$ and $s_2(\delta)$ metrics. There is not a unique best metric, which was also observed in the 3-way prediction reports of Table 3 (Bottom). Spectral embedding outperforms neural methods for contact-primary-school (unweighted $s_2$) and coauth-MAG-Geology (weighted $s_0$).

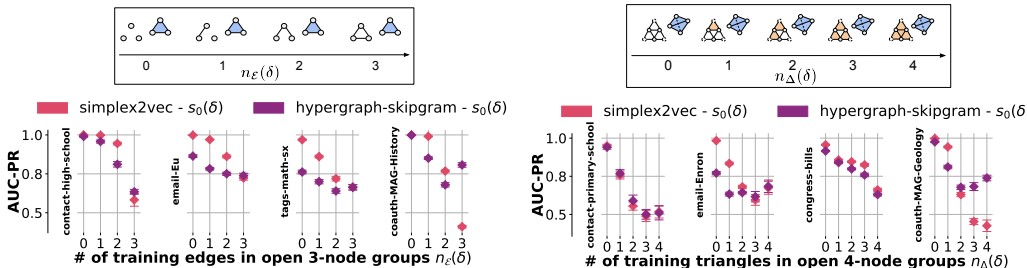

**Figure A2:** Calibrated AUC-PR scores on higher-order link reconstruction for SIMPLEX2VEC (trained on $\mathcal{H}_1$) compared with walk-based hypergraph embeddings, with similarity $s_0$. On the left are shown similarity indices varying the parameter $n_{\mathcal{E}}$ for 3-node interactions; on the right similarity indices varying the parameter $n_\Delta$ for 4-node interactions. Metrics are computed in unweighted representations. Label unbalancing in each sample is uniformly drawn between 1:1 and 1:5000.

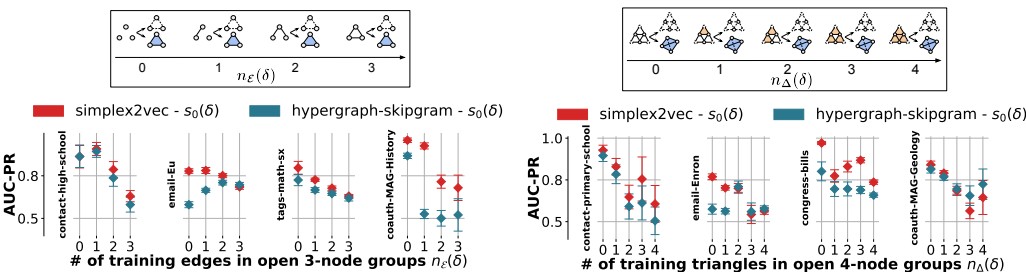

**Figure A3:** Calibrated AUC-PR scores on higher-order link prediction for SIMPLEX2VEC (trained on $\mathcal{H}_1$) compared with walk-based hypergraph embeddings, with similarity $s_0$. On the left are shown similarity indices varying the parameter $n_{\mathcal{E}}$ for 3-node interactions; on the right similarity indices varying the parameter $n_\Delta$ for 4-node interactions. Metrics are computed in unweighted representations. Label unbalancing in each sample is uniformly drawn between 1:1 and 1:5000.

## A.2   Additional Comparison with Hypergraph-based Methods

***Random Walk Encodings.*** In Figures A2 and A3 we compare classification scores respectively for reconstruction and prediction of higher-order links, among SIMPLEX2VEC and skip-gram node embeddings generated with 1st-order random walks [14] on the unweighted hypergraph structure of the input data (we use the same setup for WORD2VEC : $T = 10$, 5 epochs, 10 random walks of length 80 per node). Even SIMPLEX2VEC is trained with *Unweighted* walk transitions, leading to a similar 1st-order random walk strategy (but, on a different topological structure). The hypergraph contains hyperedges (formed by at least 2 nodes) that are simplices of $\mathcal{H}_k$, where $k = 2, 3$ is the order of simplices involved in the classification task. Even comparing node-level similarity indices, we notice that SIMPLEX2VEC outperforms hypergraph-based node embeddings in the majority of the datasets, except in the reconstruction of densely connected configurations for co-authorship data.

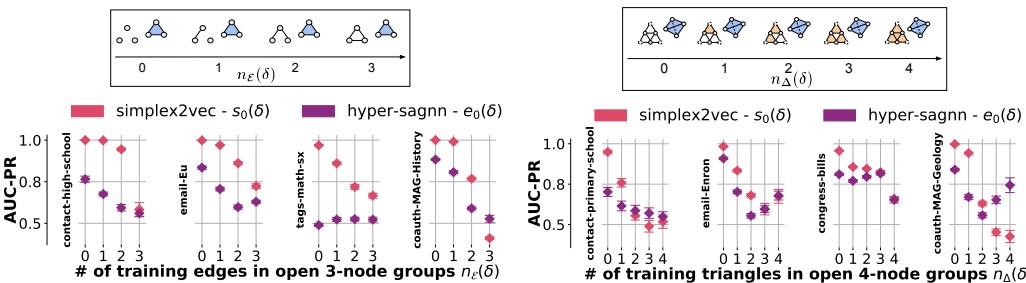

**Figure A4:** Calibrated AUC-PR scores on higher-order link reconstruction for SIMPLEX2VEC (trained on $\mathcal{H}_1$) compared with Hyper-SAGNN node embeddings. On the left are shown similarity indices varying the parameter $n_{\mathcal{E}}$ for 3-node interactions; on the right similarity indices varying the parameter $n_{\Delta}$ for 4-node interactions. Metrics are computed in unweighted representations. Label unbalancing in each sample is uniformly drawn between 1:1 and 1:5000.

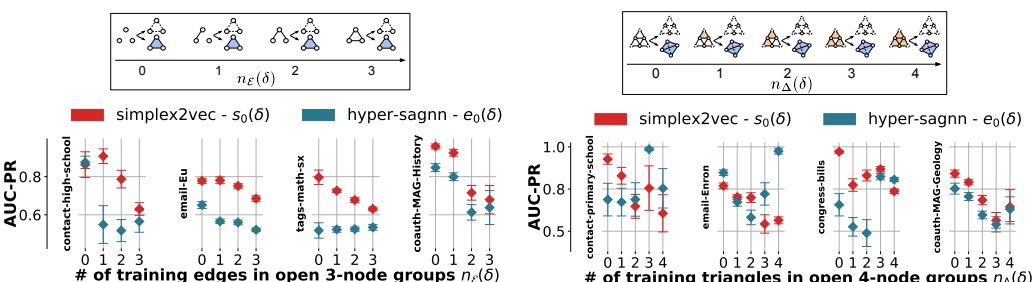

**Figure A5:** Calibrated AUC-PR scores on higher-order link prediction for SIMPLEX2VEC (trained on $\mathcal{H}_1$) compared with Hyper-SAGNN node embeddings. On the left are shown similarity indices varying the parameter $n_{\mathcal{E}}$ for 3-node interactions; on the right similarity indices varying the parameter $n_{\Delta}$ for 4-node interactions. Metrics are computed in unweighted representations. Label unbalancing in each sample is uniformly drawn between 1:1 and 1:5000.

***Hyper-SAGNN Embeddings.*** In Figures A4 and A5 we compare classification scores respectively for reconstruction and prediction of higher-order links, among SIMPLEX2VEC and Hyper-SAGNN [23] node embeddings on the unweighted hypergraph structure of the input data. Due to the model architecture, we compute hyperedge likelihood scores for Hyper-SAGNN combining embeddings with the same euclidean functional form optimized during model training, as $e_0(\delta) = \frac{1}{|\delta|} \sum_{i \in \delta} |\mathbf{d}_i - \mathbf{s}_i|^2$, where the pair $(\mathbf{s}_i, \mathbf{d}_i)$ corresponds to the (*static*, *dynamic*) embeddings of node $i$ as explained in the paper. In this setup, we notice that SIMPLEX2VEC outperforms Hyper-SAGNN embeddings in the larger part of experiments.

One of the main drawbacks of existing hypergraph-based methods (e.g., [16, 18, 23, 24]) is that they are limited to compute 0-simplex representations (node embeddings), making impossible the use of higher-order proximities (computed with interaction embeddings, like edges and triangles) similarly to the ones showed in Figures 2 and A1 (c)(d).

