# OpenReview forum: "Effective Higher-order Link Prediction and Reconstruction from Simplicial Complex Embeddings"
_logconference.io/LOG/2022/Conference — LoG 2022 Poster_

### Official Review · Reviewer_KfEJ · 2022-10-18

**Overall Score:** 8
**Confidence:** 4

**Review:**

Summary:

The paper tasks simplicial embedding methods against simplex prediction and simplex reconstruction for simplicial complexes. The prediction and reconstruction tasks are framed as binary classification problems, for which a simplex embedding method is proposed. The method, building upon the SIMPLEX2VEC method of Billings et al. [19], formulates a random walk based likelihood score to assign to candidate sets of vertices, which quantifies the probability of them forming a higher dimensional simplex.

Score & Comments:

Overall, I find the paper novel, nicely written, thorough and cohesive, thus I propose for it to be accepted. Specifically:

- The portfolio of neural embedding methods is expanded to tasks other than clustering of simplices, namely, reconstruction and prediction.
- The paper is well written, with clear statement of goals and motivation.
- Relevant literature is successfully identified and discussed, and the necessary background sufficiently presented.
- Evaluation tasks are clearly formulated.
- The experiments (and the accompanying supplementary material) are thoroughly described, aiding reproducibility, and alternatives are sufficiently explored (weighting schemes and baselines). The results support the usefulness of the proposed method.

Questions and comments:

 1. It is unclear how the temporal aspect of the datasets is handled in the simplex prediction task, since in Section 4.1 it is stated that time-stamps are disregarded during training. Is the task focused in predicting any interaction that might have happened in the time-span of the test set? If so, it would be interesting to see how the considered methods fare as the time horizon for prediction increases (i.e. AUC-PR vs time).

 2. It is somewhat unclear whether for the results of the k-SIMPLEX2VEC [18] method presented in Figures 2 and A1, $s_0$ corresponds to the standard node2vec, and $s_1$ and $s_2$ correspond to embeddings of 1- and 2-simplices, respectively, regardless of the dimension of the Hasse diagram $\mathcal{H}_1$ used for the proposed method. Is this the case, or is k-SIMPLEX2VEC also limited by the dimensionality of $\mathcal{H}_k$ for these experiments?

 3. While the proposed method generally outperforms k-SIMPLEX2VEC, could you please provide some insight on why its performance drastically degrades when $s_0$ and $n_\varepsilon(\delta)=3$ is considered, compared to that of k-SIMPLEX2VEC (Figures 2 and A1)?

 4. It might be beneficial to include some additionally commentary on why $n_\varepsilon=3$ is the hardest class of negative configurations, framing [37] in the context of simplicial complexes.

 5. Including the Hasse diagram considered (i.e. $\mathcal{H}_1$) in the caption of Figures 2 and A1 should enhance the clarity of otherwise very informative figures.

---

### Official Review · Reviewer_pgzN · 2022-10-22

**Overall Score:** 6
**Confidence:** 5

**Review:**

This paper focuses on the reconstruction and prediction of simplices in the form of classification tasks, where the likelihood of interacting groups is computed from the embedding features of a simplicial complex.

Main concerns:
* The problem stated in this paper is novel and it has not been well explored. However, the method (random walk) used in this paper is not novel.
* My main concern regarding this paper is the experiment section. While the authors claim there have been only a few works on simplicial complex, graph and hypergraph-based approaches can still be used as baselines in the experimental section.

---

### Official Review · Reviewer_hsgC · 2022-10-22

**Overall Score:** 5
**Confidence:** 4

**Review:**

Summary of contribution:
This paper presents a method for learning a low-dimensional embedding of a
simplicial complex, such that the locations of the vertices can be used for
prediction of simplex existence and classification tasks.

Strengths:
* The empircal results improve upon previous graph classification results.

Weaknesses:
* Some of the terminolgy seems to be not standard and/or does not acknowledge
    the standard terms.  For example, a "simplicial complex embedding" is
    typically called a "geometric realization" of the simplicial complex.  And,
    the definition of coface/face in lines 91-93 requires the coface/face to be
    co-dimension 1, which is not a standard requirement.  (e.g., the vertex [a],
    the edge [a,b], the triangle [b,d,a], and the tetrahedron [a,b,c,d]
    are all "faces" of the tetrahedron [a,b,c,d]).
* The title includes the term "higher-order", but the experimental part of the
    paper focuses on 2-simplices and 3-simplices are in the appendix. To me, this
    would be "low-dimensional" simplices.  So, there was a title/content
    miss-match for me.  In addition, deferring the entier "4-way analysis" to
    the appendix was a bit disappointing, as I would like to see some summary of
    the results within the main text body.
* The related work was a bit terse. For example, "node embeddings" and "GNNs"
    are differentiated, but the words used ("low-dimensional
    representations" and "vector representations", respectively) are not mutually
    exclusive.

Recommendation:
I recommend weak accept.  The paper strings together existing methods in a new
way, and has nice emperical results.

Questions to Authors: The future work "includes algorithmic approaches to tame
the scalability limits set by the combinatorial strucutre of the Hasse diagram."
Some references are given, but an explanation of why this wasn't yet tackled is
lacking. Can you specify where the technical challenge in this extension is?

Additional Feedback:
* The paragraph spacing seems to be over-ridden here, as paragraphs do not have
    space between them.
* "i.e." and "e.g." should have commas after them.
* Be consistent with title capitalization. (e.g., 3.2 is title capitalized, but
    3.3 is not).  Also, "Beyond" should be capitalized in a title.
* Well-defined should be hypenated.
* Line 69 (and elsewhere): when talking about this paper, it is best practice to
    use present tense instead of future tense.
* Line 107 (in Section 3.2) contains a forward reference to Section 4.
    Typically, forward references should be avoided (else, it can be easy to
    introduce circularl logic).
* Commas should be used after introductory clauses such as "In the Appendix"
    (line 196) and "In Figure 1 (Right)" (line 241).

Type of paper: Full paper proceedings track submission (max 9 main pages).  This
paper meets this requirement.

---

### Official Review · Reviewer_oMrP · 2022-10-22

**Overall Score:** 5
**Confidence:** 4

**Review:**

Summary: This paper proposes to invoke a simplicial complex embedding method (SIMPLEX2VEC) to obtain low-dimensional node-level representations, which are combined and fine-tuned for tasks of higher-order pattern prediction and reconstruction of simplexes through training on reduced Hasse diagrams with negative sampling.

Strength:
1) The higher-order task of link prediction and reconstruction for simplicial complexes is appropriately formulated as a binary classification problem.
2) A complete pipeline incorporating embedding-based methods is proposed and validated for higher-order tasks on simplicial complexes.
3) The proposed framework is thoroughly compared with other embedding approaches on higher-order tasks with multiple metrics and angles.

Weakness:
1) The usage of embedding-based methods is not well motivated for higher-order tasks, particularly capturing polyadic interactions.
2) Lack of comparisons with other representation learning methods,especially GNN-based models that have been largely ignored, such as [1-4].
3) The proposed method relies heavily on SIMPLEX2VEC, which brings several serious problems:
a) due to the combination explosion issue of Hasse Diagram, even in its reduced form, the scalability challenge remains, which fundamentally prohibits its application to larger complexes ($|\mathcal{K}| \leq 4$ presented in the experiments).
b) it is transductive due to the dependency of node embeddings obtained from the backbone.
4) The organization of the paper can be further improved, especially Section 3 and 4.

Questions:
1) Can the proposed pipeline be generalized from simplicial complexes to hypergraphs?
2) How will negative sampling of Hasse Diagram affect runtime complexity and prediction performance?
3) Whether the way in which the dataset is sampled/filtered in Sections 4.2 and 4.4 introduces biases that might benefit the proposed framework.
4) How is the distance T (P3 Line 111) between simplicial complexes defined?

This paper shows that simplicial complex embeddings can be effectively used for higher-order link prediction and reconstruction tasks. The authors also investigate how applying different designs of random walk sampling-based methods will affect the prediction results, such as node/edge proximity and weights. However, the proposed framework still suffers from the inherited issues of the backbone and is not inductive or generalized to larger complexes, which greatly restricted its application. Meanwhile, other representation learning methods, such as Hyper-GNNs are overlooked in related work and experimental comparisons. Overall, I recommend rejection and suggest that the authors further revise the paper and address the issues mentioned above to support the main claim and make the manuscript stronger.

[1] Yadati, Naganand, et al. "Hypergcn: A new method for training graph convolutional networks on hypergraphs." Advances in neural information processing systems 32 (2019).
[2] Feng, Yifan, et al. "Hypergraph neural networks." Proceedings of the AAAI conference on artificial intelligence. Vol. 33. No. 01. 2019.
[3] Bai, Song, Feihu Zhang, and Philip HS Torr. "Hypergraph convolution and hypergraph attention." Pattern Recognition 110 (2021): 107637.
[4] Srinivasan, Balasubramaniam, Da Zheng, and George Karypis. "Learning over Families of Sets-Hypergraph Representation Learning for Higher Order Tasks." Proceedings of the 2021 SIAM International Conference on Data Mining (SDM). Society for Industrial and Applied Mathematics, 2021.

---

### Meta-Review · Area_Chair_EMot · 2022-11-16

**Confidence:** 4
**Recommendation:** Accept

**Meta Review:**

This is a borderline case with disagreement among the reviewers.  Reviewers agree the formulation is novel and interesting but some of the engineering decisions are debatable.  Reviewer KfEJ seems to "champion" this work and highlights several positive aspects.  Overall the AC agrees that the problems considered in this paper (group-wise interaction prediction) are interesting and welcome at the conference.


Positive
* Improved graph classification performance
* New context, problem, and toolset for graph machine learning

Negative
* Possible missing comparisons to past work (see oMrP), although authors point out they're not a perfect fit and lack implementations
* Heavy reliance on SIMPLEX2VEC
* Relatively small expository concerns raised by reviewers (not a big deal)

Other comments
* The AC disregarded comments from reviewer pgzN since they were superficial and uninformative
* Please incorporate responses to reviewer KfEj in the final version of the paper

---

### Decision · Program_Chairs · 2022-11-22

Accept (Poster)